# English major students' satisfaction with ELSA Speak in English pronunciation courses

**Vi Thi Tuong Pham**[1], **Anh Tuan Pham**[2]*

**1** Foreign Language Department, Can Tho College, Can Tho City, Vietnam, **2** English Department, FPT University, Can Tho City, Vietnam

* anhpt66@fe.edu.vn

**Data Availability Statement:** All relevant data are within the manuscript.

**Funding:** The author(s) received no specific funding for this work.

## Abstract

Clear and accurate pronunciation plays an essential role in communicating effectively and significantly affects English learners' proficiency. ELSA Speak has been used in some schools due to its benefits to help learners improve English pronunciation. However, to optimize the application, a better understanding of EFL learners' satisfaction with ELSA Speak is essential. The novelty of this study is assessing how English major students were satisfied with using ELSA Speak to enhance their pronunciation and explore key factors that contributed to positive users' experience. Two hundred and five English major students at a Vietnamese higher education institution were surveyed by a 23-item questionnaire classified into four variables: perceived usefulness, perceived ease of use, confirmation, and satisfaction. The findings recorded a high level of satisfaction among participants. Moreover, although both perceived ease of use and confirmation showed a considerable impact on learners' satisfaction, confirmation was reported to be the determining antecedent of satisfaction. Perceived usefulness was directly influenced by perceived ease of use and had a stronger influence on confirmation compared to perceived ease of use. However, perceived usefulness did not have a directly significant correlation with learners' satisfaction. These results can add evidence of learners' satisfaction with using ELSA Speak, explain some elements affecting satisfaction, and suggest employing ELSA Speak in pronunciation courses with careful consideration of the content and teacher support as a facilitator.

## 1. Introduction

Pronunciation plays an important role in English learning since it significantly impacts learners' ability to communicate effectively [1]. Clear and accurate pronunciation is crucial for both being understood by other people and comprehending the others' spoken language. Conversely, poor pronunciation can hinder communication by causing misunderstanding [2]. However, mastering pronunciation requires not only a thorough understanding of the English sound system and phonology but also consistent practice [3]. Hence, it is pivotal to seek and propose strategies that facilitate students' English pronunciation learning.

In Vietnam, teaching pronunciation is integrated with other skills in most lessons except in English classes where dedicated pronunciation subjects have been included. Instructors in

**Competing interests:** The authors have declared that no competing interests exist.

these specialized courses tend to rely on traditional approaches and activities such as repetition, reading aloud, and playing games [4]. Hence, the appearance of ELSA Speak, a mobile app using Artificial Intelligence (AI) and speech recognition technology, marks a significant innovation in teaching pronunciation. Besides, this application has been reported to be effectively improve learners' engagement with practicing pronunciation as well as enhancing their English pronunciation skills [5–8].

Numerous studies have been conducted to find effective ways to optimize students' pronunciation learning and stimulate their motivation to engage in more regular practice. However, few studies on English major students' satisfaction with ELSA Speak, as a tool for improving pronunciation, have been found. Meanwhile, exploring students' satisfaction is essential for educational institutions and educators to gain a deeper understanding of their students' needs and concerns [9]. This can serve as a basis for proposing appropriate adjustments to the curriculum or teaching methods to help students improve their performances [10] and increase their motivation in practicing pronunciation. Hence, the current study was conducted to figure out how English major students were satisfied with ELSA Speak to enhance their pronunciation and the factors that influence their satisfaction.

## 2. Literature review

### 2.1 Features of English pronunciation

According to [11], features of English pronunciation are classified into two levels: the segmental features and suprasegment features.

As shown in Fig 1, suprasegmental features refer to sounds on a larger level that involve linking (the way of connecting the final sound of one word with the initial sound of the following word), intonation (the pitch variation when speaking), and stress that includes sentence stress (the emphasis placed on certain words in an utterance) or word stress (the prominence given to a specific syllable in a word).

Segmental features consist of components of the sounds at the micro level which are called phonemes. Phonemes comprise consonant and vowel sounds. Consonant sounds can be voiced (produced by vibrating the vocal cords) or unvoiced (made without the vibration of the vocal cords). Vowel sounds can be pronounced as single sounds (long or short sounds) or diphthongs (putting two vowel sounds together).

### 2.2 Factors affecting learning English pronunciation

Even in a homogenous class, there can be significant discrepancies in learners' English pronunciation abilities due to various factors. According to [12], important factors that influence learning pronunciation of English language students could be classified into two groups: internal and external factors. Internal factors consist of age, brain, innate ability, learners' types, and objectives. External elements encompass first language influence, motivation, teachers' types and instructional methods, the setting of the classroom, and institutional factors.

Internal factors affect learners in various ways. Research indicates that young learners are more likely to acquire better English pronunciation. At an early age, a child's brain has more plasticity than an adult's brain. That can make acquiring native-like pronunciation less challenging for a child than for an adult. Aptitude can be understood as an innate ability that enables an individual to do tasks better than others. Additionally, it was shown that learners who paid more attention to pronunciation performed better in English pronunciation. Regarding learners' types, it was noted that extrovert learners tended to have more accurate pronunciation than introvert students since they were more inclined towards taking opportunities to practice it with the target language. In terms of goals, [12] it suggested that intelligible

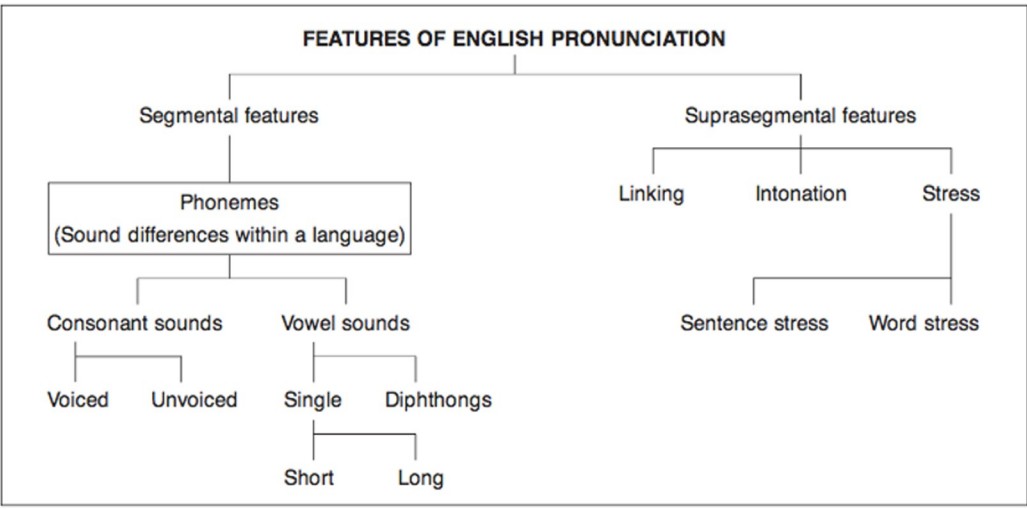

**Fig 1. Features of English pronunciation.**

pronunciation is essential for learners to bear in mind instead of trying to achieve native-like pronunciation with perfect American or British accents.

External factors are similar to internal factors in that they both encompass a variety of elements. First, the variations in phonological rules between the first language and the target language could hinder learners from mastering the target language's pronunciation [13]. On the contrary, the similar features between the two languages can make it simpler for learners to acquire the target language's pronunciation. Second, if students' orientation becomes their goal in learning pronunciation, it can help learners become more proficient. Third, teachers play an essential role in facilitating and motivating learners during the learning process. They must possess appropriate knowledge of the phonetics and phonology of the target language, select suitable classroom activities, use effective teaching methods, provide motivational feedback, and evaluate students' progress. Those things can shape learners' most favorable performances. Finally, creating more opportunities for learners to practice English not only in the in-class environment but also providing enough out-of-class activities for students to be exposed to substitute real situations, is imperative because learners can get access to a wider range of pronunciation varieties [14].

## 2.3 Teaching pronunciation with technological tools

Applying technology tools in teaching and learning pronunciation has been popular for many years. However, choosing the best tools and using them needs to meet some requirements. According to [15], some criteria should be considered to select the best tools to teach pronunciation: suitability to the learning goals, quality and precision, practicality of use, and affordability. Appropriateness to learning objectives means that the tools can help teachers complete teaching tasks and learners achieve their goals in learning pronunciation. Quality and accuracy refer to the accurate information about principles of phonology and pedagogy that the tools offer. Practicality of use depicts whether the tools are easy to learn or use and whether their functions are performed accurately and smoothly. Cost is related to the reasonable fee users have to pay to use the tools.

Following are some common pedagogical tasks required in the process of teaching pronunciation with technology tools [16]. The first one is providing a pronunciation model in which

examples of pronouncing individual sounds and words or longer speech are given to learners. The next task is recording learners' pronunciation practice. Learners can enhance their self-correction ability by listening to their pronunciation in recordings as well as increase their motivation and interest. Another task is collecting and giving feedback on students' recorded practice. With the use of technology tools, this process could be simpler and more effective. The final task is providing independent pronunciation practice with a wide range of exercises with different levels of difficulty that enable learners to practice independently.

### 2.4 ELSA Speak application

**2.4.1 Features of ELSA Speak application.** ELSA Speak is an educational app developed using AI and speech recognition technology. It can be easily installed on iPads, Android tablets, and smartphones that run on Android/iOS. Users need to purchase an account to access it. ELSA Speak provides a wide range of activities across various skills, levels, topics, and other features. With this application, users can study anytime and anywhere they like with an Internet connection. In addition, they can listen to models, practice pronouncing words, sentences, or conversations, and record their pronunciation by themselves. Then, they can receive instant feedback on their pronunciation mistakes, specific suggestions to improve them, and reports on how much progress they have had after a certain time of practice. Moreover, users can set up their goals for practicing with casual practice (10 minutes a day), standard practice (15 minutes a day), or serious practice (20 minutes a day), and choose the display language such as English, Vietnamese, or Chinese. Additionally, its content comprises free tests for users to customize their learning paths, a dictionary to check pronunciation, different exercises for practicing pronunciation and listening related to segmental and suprasegmental features such as vowels, consonant clusters, linking sound, stress, intonation, and so on. Users can choose exercises that match their proficiency levels. Besides the available content, ELSA Speak will update the new lessons regularly with multiple topics, or users can suggest topics they would like to practice by contacting the developers. Fig 2 presents some displays of ELSA Speak.

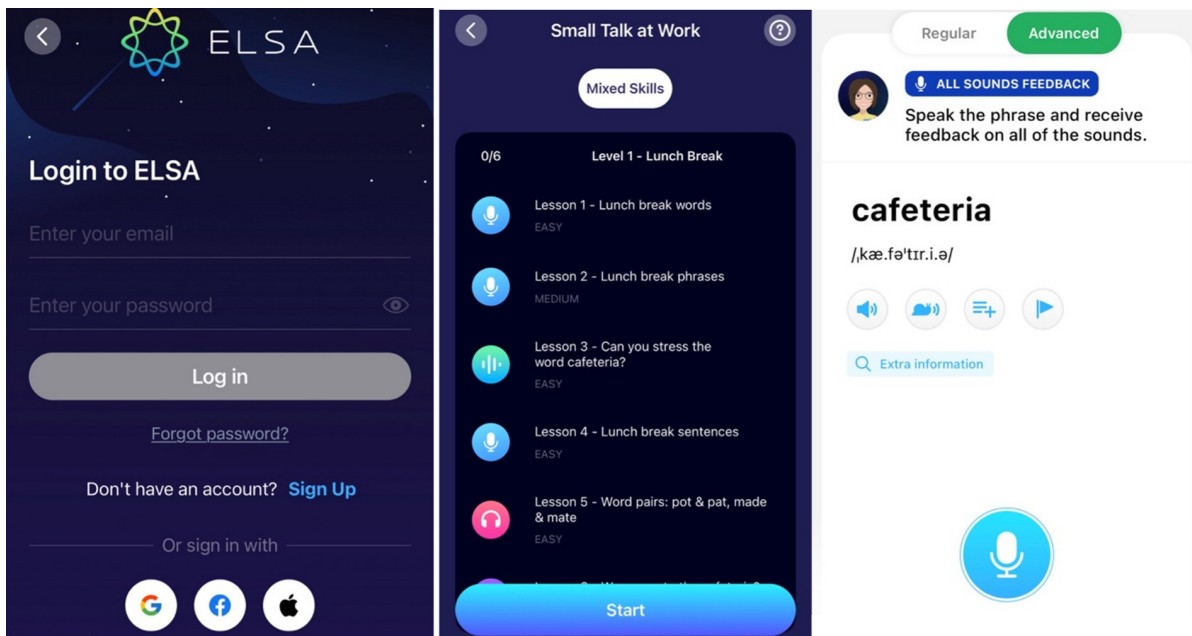

**Fig 2. Screenshots of ELSA Speak application.**

All of the above features show that ELSA Speak could be a potential application to increase the efficacy of teaching and learning pronunciation and enhance learners' motivation and satisfaction as well.

**2.4.2 Obstacles to using ELSA Speak in pronunciation.** When applying ELSA Speak application to improve English pronunciation, several challenges can hinder students' learning process. Completing exercises that were repeated made students uncomfortable and disappointed [17]. Moreover, technical problems such as unstable internet connection, too bright screens, and the application's limitations in voice recognition can prevent accurate feedback from being consistently delivered [18,19]. Besides, the lack of direct human interaction is a popular limitation that can cause problems for students who favor more interactive learning experiences. Next, ELSA Speak as other AI language learning tools may struggle to capture the language's cultural and contextual nuances including idioms, regional accents, or slang. This can result in misunderstanding, especially when dealing with more technical language. Finally, the extensive data that the app needs to rely on can pose challenges for underrepresented languages or dialects. Hence, this may lead to limited resources or biased training materials for those languages [19].

## 2.5 Related studies

A study was conducted on 30 students who studied the Easy English course at a basic level to enhance learners' pronunciation skills by using ELSA Speak. Data were collected through a questionnaire, observations, a pronunciation test, and interviews. The findings indicated that students' English pronunciation skills and motivation benefited from using ELSA Speak [6]. These findings echoed several studies that highlighted the efficiency of applying ELSA Speak to increase learners' pronunciation [1,5,7,20].

Regarding utilizing ELSA to enhance Business English communication, a quasi-experimental study was conducted on ninety-nine management students from a business school. The results revealed that students' productive competency, receptive competency, lexicon usage, phonetics utterances, and grammar skills were significantly benefited from using ELSA as an Education 4.0 tool. However, listening skills were mostly improved compared to other skills, and speaking skills showed greater development than writing in terms of productive competency [21].

To examine learners' perception of utilizing ELSA Speak in learning pronunciation at the tertiary education level, a questionnaire with 41 items and two short-answer questions was delivered to 180 students who studied the high-quality program at Can Tho University. The findings stated that ELSA Speak was perceived positively after being used for six months in an English pronunciation practice course [17]. These findings were consistent with another research that was carried out on 112 learners in the English Department at an Indonesian university [8].

It can be seen that ELSA Speak has been proven to have a predominantly positive reception among learners at different levels, enhance learners' English competence and skills in various aspects, and improve their ability to pronounce English in particular.

## 2.6 Theoretical frameworks and research hypotheses

**2.6.1 Technology Acceptance Model (TAM).** This model was based on the Theory of Reasoned Action and sheds light on what underpinned individuals' willingness to accept new technology [22]. According to TAM, the acceptance of innovative technology processes is triggered when users believe that the application is user-friendly and useful for them based on the features of its system design. These beliefs motivate users to adopt the application. Finally,

these expectations urge the users to use the application intentionally (see Fig 3). Notably, perceived ease of use has an essential role in the model since it not only supports the perceived usefulness but also forms the determination to use an app. In other words, it can be implied that if an application is believed to be easy to use, it is possibly perceived as useful for the users. As a result, users are more likely to adopt the target system [22,23].

**2.6.2 Expectation-Confirmation Theory (ECT).** This theory originally appeared in the psychology and marketing field to explain how customers' expectation in the pre-consumption stage affects their thoughts and actions related to a product or service in the post-consumption stage [24]. However, it has been increasingly used in other fields; for example, in information technology, to anticipate the continuance of using electronic commerce services [25], identify factors of continued use of the Word Wide Web [26], measure the satisfaction of the web users [27], or in medical education to figure out dominant elements affecting learners' satisfaction and e-learning continued intention [28].

As can be seen from Fig 4, the initial expectation before purchasing, which can be formed by customers' knowledge and former experience about the product or service, plays a role as criteria that customers base on to evaluate their satisfaction. Besides, it shapes their confirmation, which is the customers' view on how well the actual effectiveness that the product or service is compared to their previous expectations. Confirmation can be positive when perceived performance surpasses the prior expectation, simply when it meets the expectation exactly, or negative when the perceived performance goes below the expectation.

Next, customers can show different levels of satisfaction based on their confirmation. Positive and simple confirmation can lead to a sense of satisfaction by fostering customers' favorable attitudes toward the products or services. In contrast, dissatisfaction can be generated by negative confirmation. Finally, repurchasing intention can be determined by customers' satisfaction with their experience when using that product or service.

**2.6.3 Proposed research model.** Drawing on TAM theory, users' perceived ease of use and perceived usefulness are considered very important variables in predicting learners' behavior since they form the expectations leading to the users' affective responses - the attitude toward applying the application or intention to use it, which ultimately affect users' actual use

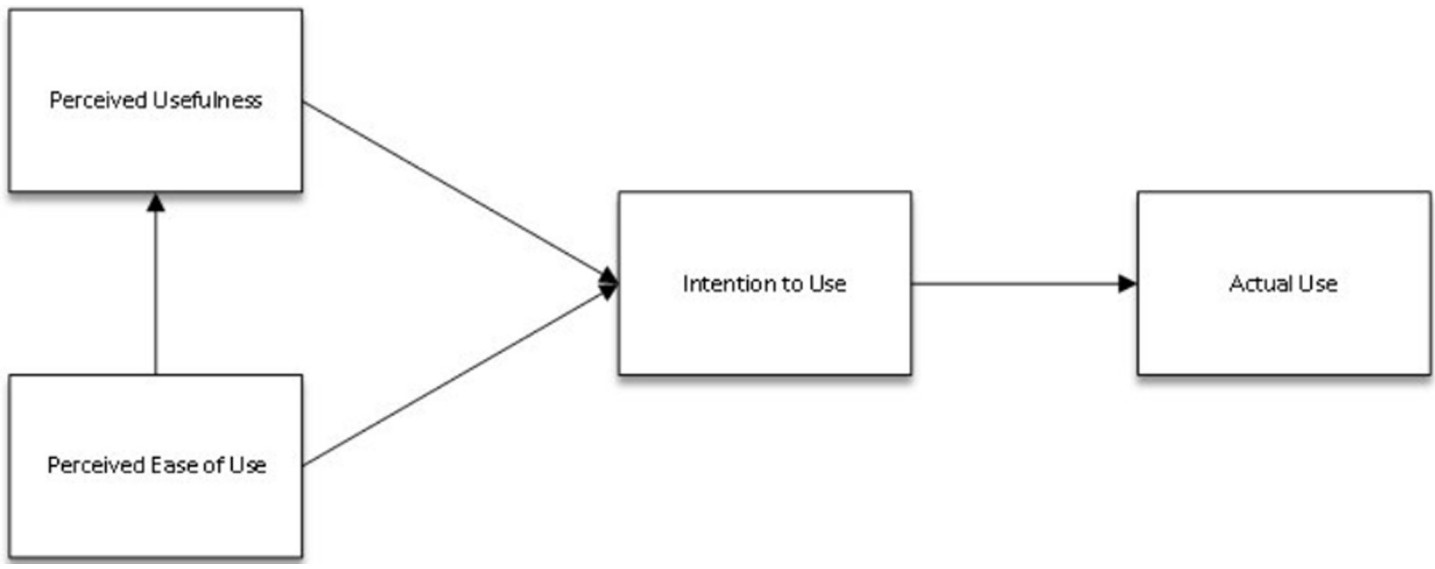

**Fig 3. Technology acceptance model.**

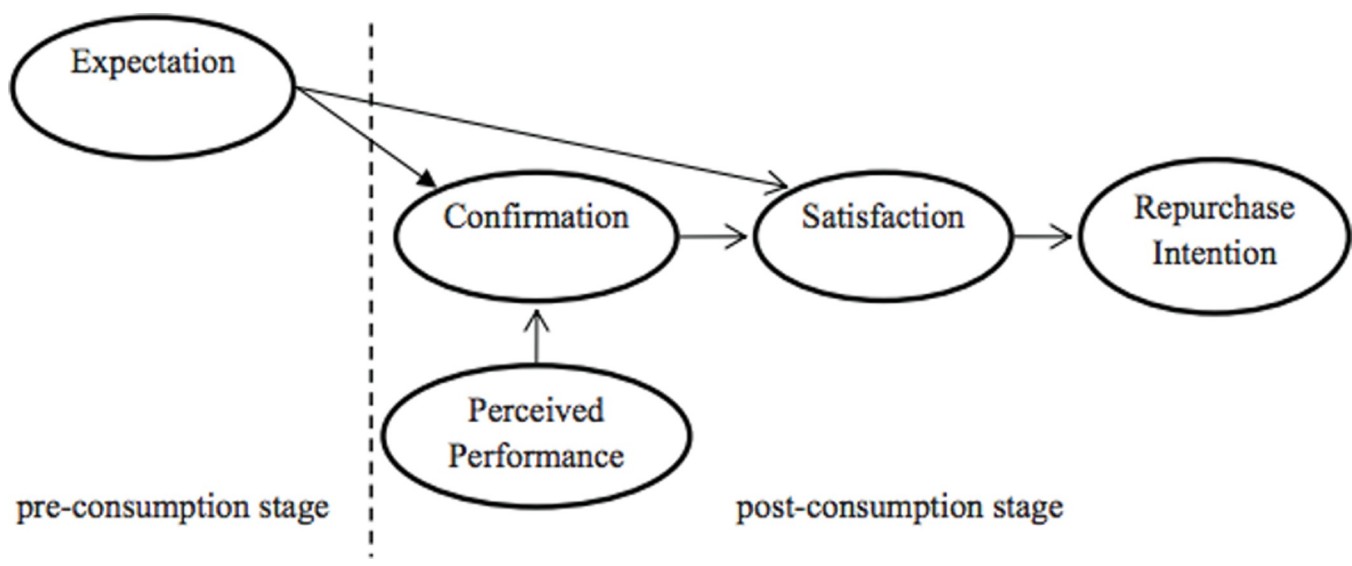

**Fig 4. The expectation confirmation theory.**

behavior [22,23]. Besides, according to ECT theory, the expectations before using a product play an essential role in users' confirmation because they set up criteria for users to examine how effective the products can be. Then, it is noted that confirmation is a major antecedent of users' satisfaction [24]. Building on the two theories above, the proposed research model comprises four constructs: perceived usefulness, perceived ease of use, satisfaction, and confirmation. Table 1 presents the constructs and their definitions within the framework of this study.

Guided by the combination of TAM and ECT and the theories above, Fig 5 shows the hypothesized relationships among these constructs. Specifically, the hypotheses suggested in this study were:

H1: Students' perceived ease of use is directly related to their perceived usefulness of ELSA Speak.

H2: Students' perceived ease of use is directly related to their confirmation of ELSA Speak.

H3: Students' perceived usefulness of ELSA Speak is directly related to their confirmation of ELSA Speak.

H4: Students' perceived ease of use is directly related to learners' satisfaction of Elsa Speak

H5: Students' confirmation is directly related to learners' satisfaction of Elsa Speak

H6: Perceived usefulness is directly related to learners' satisfaction of Elsa Speak

**Table 1. Constructs in the proposed research model and their definitions.**

| Constructs | Definitions |
| --- | --- |
| Perceived usefulness | Learners' belief that ELSA Speak can improve their English pronunciation performance |
| Perceived ease of use | Learners' belief that ELSA Speak can require free of effort |
| Confirmation | Learners' perception about ELSA Speak after verifying their belief about ELSA Speak and its actual performance |
| Satisfaction | Learners' pleasant feeling when their desire or need fulfilled |

According to Fig 5, the first hypothesis depicts that if students find ELSA Speak easy to use, they are likely to believe it as useful. This is in line with the key idea of TAM, in which the ease of use is a significant predictor of perceived usefulness. The second hypothesis suggests that perceiving ELSA Speak as easy to use can urge students to confirm its effectiveness in improving their English pronunciation. This aligns with ECT, where positive experiences lead to the confirmation of expectations. The third hypothesis posits that if students who perceive ELSA Speak as a useful tool are more inclined to affirm its effectiveness. This is a core idea of ECT, which describes that confirmation is led by perceived usefulness. The fourth hypothesis posits that students tend to be satisfied with ELSA Speak if they find it easy to use, aligning with TAM, which emphasizes the ease of use as a predictor of user's satisfaction. The fifth hypothesis proposes that when students validate the effectiveness of ELSA Speak, they are more likely to satisfy with it. This is consistent with ECT, where users' satisfaction rises when positive expectations are confirmed. The sixth hypothesis suggests that if students perceive ELSA Speak as useful, they are likely to feel satisfied with it. This reflects the core idea of TAM, where perceived usefulness is considered as a main factor influencing satisfaction.

## 2.7 Significance of the study and research questions

ELSA Speak is a widely recognized tool employed to facilitate learners in learning English pronunciation. The current study provides novel insights into the application of ELSA Speak mobile application by assessing how English major students rated their satisfaction with using ELSA Speak for enhancing English pronunciation and identifying what factors affected their satisfaction by finding out the answers to the following questions.

1. How did students rate their overall satisfaction with using ELSA Speak to improve their English pronunciation?

2. What factors influenced students' satisfaction with utilizing ELSA Speak?

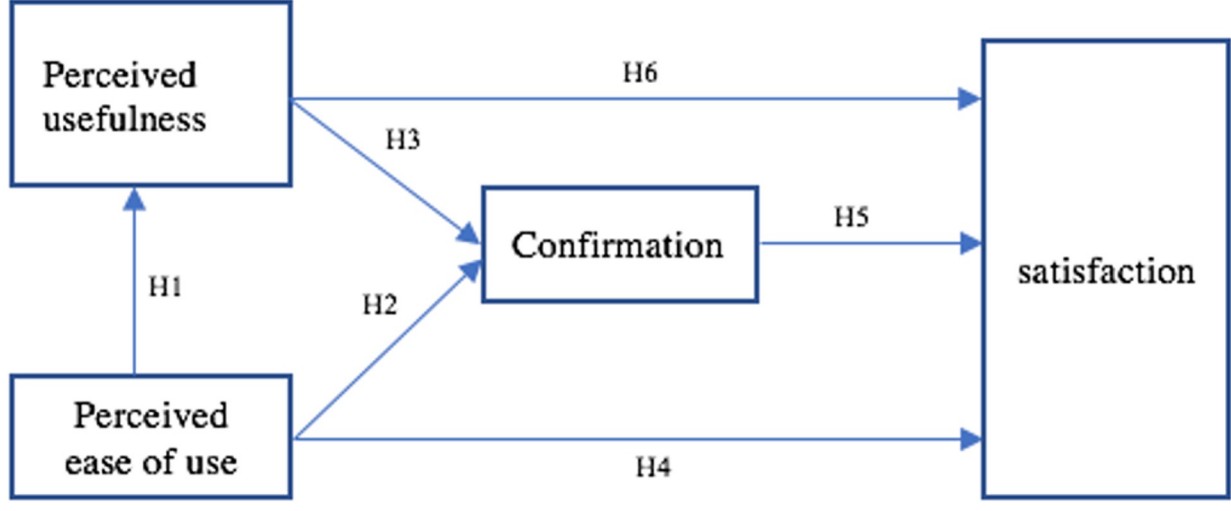

**Fig 5. Proposed research model and hypotheses.**

## 3. Methodology

### 3.1 Research design

This study utilizes a descriptive and correlational research design to analyze quantitative data. According to [29], one of the purposes of descriptive research is to systematize and depict precisely the facts and characteristics of a specific population and identify relationships among variables. This approach is suitable for the current study's objectives, which include assessing English major students' satisfaction with ELSA Speak for improving their English pronunciation and examining the relationships among constructs such as perceived ease of use, perceived usefulness, confirmation and satisfaction.

### 3.2 Participants and context

This study employed a convenience sampling technique to select the participants. Two hundred and five English major students involved in the study were from a higher education institution in Can Tho city in Vietnam. Table 2 shows that all students were aged between 19 and 21. Around 46.83% of the participants were first-year students. Female students comprised 77.56% of the sample, outnumbering male students at 22.44%.

The ethics approval was obtained from FPT University with reference No. 20240608.02. Informed written consent was obtained from participants by a self-administered online questionnaire. Participants were promised that their data would only be used for research reasons, and confidentiality was maintained throughout the research. The individual had no limitations on their right to withdraw from the study.

All participants used ELSA Speak during a 15-week English Pronunciation course at the university. Before starting the course, students were introduced to ELSA Speak in terms of its interface and usage. Then, in the course, each student was given an account to get access to assignments that were designed based on the English Pronunciation course's curriculum. Students were required to complete three or four assignments on ELSA Speak every week. The teachers tracked and monitored their students' practice and achievement to give necessary suggestions weekly. At the end of the course, students' achievements using ELSA Speak were recorded as part of their course results.

### 3.3 Instruments

In this research, a 5-point Likert scale questionnaire was designed by the researcher to investigate learners' satisfaction with ELSA Speak to improve their English pronunciation. Before putting it into use, the questionnaire was given out to two English instructors to check the accuracy and clarify mistakes. The satisfactory questionnaire consisted of 23 items that were grouped into four categories: perceived usefulness (7 items), perceived ease of use (3 items), confirmation (10 items), and satisfaction (3 items). The result for Cronbach's Alpha value was .964.

**Table 2. Demographic information about participants.**

| Year of study | Number of students (%) | Age | Gender | |
|---|---|---|---|---|
| | | | **No. of males** | **No. of females** |
| First year | 96 (46.83%) | 18–21 | 20 | 76 |
| Second year | 109 (53.17%) | 19–21 | 26 | 83 |
| Total | 205 | | 46 (22.44%) | 159 (77.56%) |

### 3.4 Data collection procedure and analysis

The questionnaires were delivered to the participants in print between August 15 and September 14, 2024. Written informed consent was obtained before the data collection process. Before completing the questionnaire, the requirements were explained to the participants, and any ambiguous terms were clarified. The data collected was then analyzed by using SPSS version 25 and AMOS 24. Exploratory Factor Analysis (EFA) was applied to determine the underlying latent factors. Confirmatory Factor Analysis (CFA) was used to test the validity of the structure of the questionnaire. Structural Equation Modeling (SEM) was employed to examine the relationships between factors and their indicators as well as the direct effects between these factors in the research model.

## 4. Results

### 4.1 Learners' satisfaction with ELSA Speak to enhance their English pronunciation

As reported in Table 3, the average value of learners' satisfaction with ELSA Speak is high ($M$ = 4.112, $SD$ = .806). Besides, one sample t-test was run to check whether a significant difference between the mean of learners' satisfaction in a statistics lesson and in the true population existed. The results showed that the mean of learners' satisfaction in a statistics lesson ($M$ = 4.112, $SD$ = .806) was significantly different in the true population $[t(204) = 1.994, p = .048]$. Therefore, it can be said that many participants were satisfied with ELSA Speak to enhance their English pronunciation.

### 4.2 Factors affecting learners' satisfaction with ELSA Speak to enhance English pronunciation

**4.2.1 The reliability of the research instruments.** Table 4 shows the results of Cronbach's Alpha for all constructs were satisfactory ($\alpha$>.7). Moreover, the correlation coefficients of all items were higher than .3. Those results proved that these variables satisfied the standard Cronbach's Alpha performance of the analysis and they were reliable for further analysis [30].

**4.2.2 Exploratory factor analysis (EFA).** EFA was used to examine the relationships among variables and remove insignificant items in the questionnaire. The coefficient Kaiser-Meyer-Olkin Measure of Sampling Adequacy is .949 which is greater than .5, and Barlett's Test of sphericity was significant (p = .000 < .05) proving that the observed variables were correlated with the components.

As can be seen from Table 5, four components had Total Initial Eigenvalues greater than 1 and they explained 72.821% of the variance (>50%). Hence, the proposed research model including these four components would be satisfactory for further analysis.

Besides, the factor loadings, as presented in Table 6, all weighed greater than .5 showing that the observed variables had a strong correlation with the constructs and had good quality [31]. Besides, the values of Composite Reliability (CR) of all constructs exceeded .6. indicating the reliability and internal consistency of all latent constructs in the proposed research model. Moreover, the Average Variance Extracted (AVE) of each construct was greater than .5. It means the average percentage explained by the observed items for each latent construct was

**Table 3. Learners' satisfaction with ELSA Speak to enhance English pronunciation.**

|  | N | Min | Max | Mean | SD |
|---|---|---|---|---|---|
| Learners' satisfaction | 205 | 1.000 | 5.000 | 4.112 | .806 |

**Table 4. The results of Cronbach's Alpha.**

| Constructs | Code | Number of Items | Cronbach's Alpha | Item-total correlation |
|---|---|---|---|---|
| Perceived usefulness | PU | 7 | .917 | .576 - .811 |
| Perceived ease of use | PEU | 3 | .838 | .636 - .802 |
| Confirmation | C | 10 | .954 | .672 - .802 |
| Satisfaction | S | 3 | .870 | .687 - .790 |

greater than 50%. Therefore, the proposed research model is reliable for measuring the intended latent constructs.

**4.2.3 The fitness of the proposed research model.** The fitness indexes for the measurement model, shown in Table 7, were all within acceptable thresholds, suggesting a good fit for the proposed model (CMIN/df = 2.147, CFI = .936, GFI = .838, TLI = .927, RMSEA = .075) [32–34].

Table 8 shows indices measuring the discriminant validity of the measurement model. Accordingly, the maximum shared variance (MSV) is lower than the average variance extracted for all variables. Additionally, the square root of the average variance extracted (AVE) exceeds all inter-construct correlations. Therefore, the proposed model has ensured the discriminant validity [32].

**4.2.4 Hypothesis testing.** Fig 6 shows that the results from the structural equation modeling (SEM) method indicates multiple fit indices of the structural model. The Chi-square/df value was 2.147 less than 5. The comparative fit index (CFI) value was .936 which was greater than .9. The goodness of fit index (GFI) was .838, greater than .8. Tucker-Lewis index was .927, greater than .9. Finally, the root square error of approximation (RMSEA) was .075, which was less than .08 [32–34] (Fig 6).

As reported in Table 9, all hypotheses were supported because their Sig values were less than .05. First, perceived ease of use positively influenced and weighed 82.5% with the variation of learners' perceived usefulness of ELSA Speak. Both perceived usefulness and ease of use were reported to have a direct relationship with students' confirmation; however, the perceived usefulness showed a substantially larger effect size compared to the perceived ease of use ($\beta$ = .574, $\beta$ = .306 respectively). Besides, confirmation after using ELSA speak drove learners' satisfaction because it directly had a greater effect on learners' satisfaction ($\beta$ = .657) compared to perceived ease of use ($\beta$ = .369). Meanwhile, perceived usefulness did not show a direct impact on learners' satisfaction ($p$ = .127 < .05).

# 5. Discussion

The results of the current study revealed that many learners were highly satisfied with ELSA Speak to enhance their English pronunciation. This is consistent with the findings of previous studies that highlighted learners' high rate of satisfaction or positive perception toward employing the ELSA Speak app in learning English pronunciation [17,20,35]. This could be

**Table 5. Total variance explained.**

| Components | Initial Eigenvalues | | |
|---|---|---|---|
| | Total | % of Variance | Cumulative % |
| 1 | 14.087 | 56.348 | 56.348 |
| 2 | 1.728 | 6.912 | 63.260 |
| 3 | 1.388 | 5.550 | 68.810 |
| 4 | 1.003 | 4.011 | 72.821 |

**Table 6. Exploratory factor analysis.**

| Constructs | Items | Factor Loadings | CR | AVE |
|---|---|---|---|---|
| Perceived usefulness (PU) | 7 | .542 - .808 | .914 | .662 |
| Perceived ease of use (PEU) | 3 | .627 - .803 | .835 | .625 |
| Confirmation (C) | 10 | .647 - .800 | .954 | .689 |
| Satisfaction (S) | 3 | .741 - .800 | .873 | .678 |

Note: CR = Composite Reliability; AVE = Average Variance Extracted

attributed to the various useful features of ELSA Speak. Notably, the opportunity for learners to practice pronunciation independently and receive immediate, accurate were the key factors in student satisfaction [7,8]. Then, the ELSA Speak app has been noted to benefit learners in improving their pronunciation due to including a large variety of exercises for practice with different levels [36]. In addition, learners can easily practice with exercises of appropriate difficulty to their level and study at their own pace. Besides, thanks to the combination of speech recognition technology and advanced AI implementation, this app can give learners appropriate feedback after comparing their pronunciation and the standard accent of native speakers which is advantageous to students' learning [37,38].

The finding showed a strong correlation between students' belief in the ease of use and their perceived usefulness of ELSA Speak. This result was similar to TAM theory that users' perceived ease of use has directly affected their perceived usefulness [22,39]. Regarding the current research, it was the first time the participants experienced using ELSA Speak. Therefore, ease of use is crucial in encouraging students to try the app, as they are more likely to explore and recognize its usefulness if the technology is easy to use [40]. Additionally, ELSA Speak has been reviewed as a user-friendly application. For example, it empowers users to opt for their native language which makes instruction and feedback clearer and more understandable [38]. It not only creates the students' first positive impressions of this application but also enables learners to complete the tasks more effectively.

The results presented that although both learners' perceived ease of use and usefulness positively affected their confirmation after using ELSA Speak, students' belief in the usefulness of this app was found to be a more determining antecedent. This was similar to several previous studies that emphasized the more significant role of perceived usefulness in predicting users' behavior due to its direct impact on the actual use or confirmation compared to perceived ease of use [23,28]. Learners could base their perception of the degree to which using ELSA Speak can improve their outcomes, as well as the belief that using this app requires free effort to establish a set of standards to compare with its actual performance after using ELSA Speak. However, enhancing learners' English pronunciation is the main goal for employing ELSA

**Table 7. CFA measurement model fit indices.**

| | Observed value | Acceptable Index Value | Results |
|---|---|---|---|
| CMIN/df | 2.147 | < 5 | Acceptable |
| GFI | .838 | ≥ .8 [*] | Acceptable |
| CFI | .936 | > .9 | Acceptable |
| TLI | .927 | > .9 | Acceptable |
| RMSEA | .075 | < .08 | Acceptable |

* [33,34].

**Table 8. Discriminant validity measures.**

|     | AVE  | MSV  | MaxR(H) | PU       | C        | S       | PEU   |
|-----|------|------|---------|----------|----------|---------|-------|
| PU  | .662 | .601 | .915    | **.814** |          |         |       |
| C   | .689 | .620 | .932    | .629***  | **.830** |         |       |
| S   | .678 | .511 | .897    | .592**   | .705***  | **.823**|       |
| PEU | .625 | .620 | .901    | .775***  | .788***  | 0.715** | **.790** |

Speak in pronunciation classes [1,5,8,20,38]. Hence, the role of perceived usefulness is still dominant.

The findings also suggested that participants' confirmation was the major antecedent to predicting students' satisfaction. This finding aligns with Expectation-Confirmation Theory [24] and in other studies that emphasize the positive impact of learners' confirmation on their satisfaction [28,41]. It could be explained that confirmation is directly related to students' expectations as well as subjective evaluation of how well a tool or a system performs after using it [42]. Therefore, it can be said that its essential role in creating learners' satisfaction has been well established.

Finally, the results revealed that while students' perceived ease of use directly affected their satisfaction after using ELSA Speak, no significant correlation was found between learners' perceived usefulness and their satisfaction levels. This finding was not consistent with several studies which highlighted the positive impact of perceived ease of use and perceived usefulness on the users' satisfaction [25,38,43,44] or only perceived usefulness was stated as a strong predictor of students' satisfaction [28]. However, the finding was supported by [45], showing no significant association between learners' perceived usefulness and their satisfaction. This can be because nowadays learners tend to choose e-learning tools with more familiar interfaces [46].

## 6. Implications, limitations and recommendations

The current study aims to examine how English major students were satisfied with ELSA Speak to improve their English pronunciation and figure out what factors influence learners' satisfaction with this application in pronunciation classes. This research model provides a

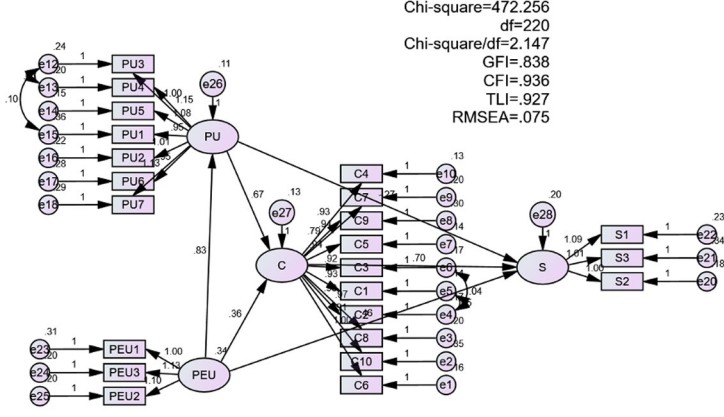

**Fig 6. Effects on students' satisfaction with ELSA Speak to improve English pronunciation.**

**Table 9. Results of the structural equation model.**

| | Description of hypothesis | Significant results (p-value) | Standardized regression coefficients | Hypotheses results |
|---|---|---|---|---|
| H1 | Students' perceived ease of use is directly related to their perceived usefulness of ELSA Speak. | *** | .825 | supported |
| H2 | Students' perceived ease of use is directly related to their confirmation of ELSA Speak. | *.004* | .306 | supported |
| H3 | Students' perceived usefulness of ELSA Speak is directly related to their confirmation of ELSA Speak. | *** | .574 | supported |
| H4 | Students' perceived ease of use is directly related to learners' satisfaction of ELSA Speak | *.006* | .369 | supported |
| H5 | Students' confirmation is directly related to learners' satisfaction of ELSA Speak | *** | .657 | Supported |
| H6 | Perceived usefulness is directly related to learners' satisfaction of using ELSA Speak | *.127* | -.215 | Not supported |

Notes: ***, p-value < .001. Significant at the .05 level.

comprehensive understanding of factors affecting students' satisfaction with using ELSA Speak in learning pronunciation. By including perceived ease of use, perceived usefulness, and confirmation, it offers a robust framework for assessing the app's effectiveness. Perceived ease of use emphasizes the significance of a user-friendly interface and clear instructions. Perceived usefulness focuses on the value that learners put on the app's features. Confirmation which is based on users' positive reviews reinforces the credibility and effectiveness of ELSA Speak. That can boost students' satisfaction. This model enables researchers to delve into factors that lead to users' positive experiences and guide future enhancement in language learning technology.

Drawing on the findings, the current study recommends several practical implications. First, the study pointed out participants' great satisfaction with ELSA Speak after employing it in a 15-week course in English pronunciation. This result suggests that curriculum designers at educational institutions should consider including ELSA Speak as an implementation for pronunciation teaching curriculum due to its benefits in boosting learners' pronunciation. However, it is necessary to design or select the content that is appropriate for the objectives of the course as well as the student's level [15]. Practitioners in the field are also encouraged to adopt ELSA Speak to enhance their English pronunciation because of the available features offered by this app, which can get users more engaged in practicing pronunciation [7].

Second, the results emphasized a great correlation between learners' perceived ease of use and perceived usefulness, and both of them had a direct effect on learners' confirmation although perceived usefulness showed a stronger effect. Moreover, a direct relation between perceived ease of use and learners' satisfaction was confirmed. Hence, it is for teachers and educators themselves to discover ELSA Speak to understand clearly its instructions and features, then clarify how to use this application to make it easier for students. Moreover, instructors can assign personalized exercises to cater to students' various learning styles and levels. Besides, the program designers should consider making ELSA's interface simpler and more clearly, improving the quality of voice recognition technology to allow learners to practice pronunciation without frustration, and offering a wide range of learning lessons or quizzes to motivate students to engage more in ELSA Speak's learning activities. This can create learners' belief that using this application will be easy and useful for them, which stimulates them to experience using it [39].

Finally, the finding revealed learners' confirmation as a major predictor of learners' satisfaction. Therefore, teachers are expected to demonstrate their roles as facilitators [47]. In other

words, teachers and learners are advised to discuss with each other about difficulties in the learning process to find out the best solutions. Learners need to keep track of their improvement based on the app's metrics and feedback. That not only motivates them but also helps them to make necessary adjustments during their learning process. Then, positive experiences, or learning tips with ELSA Speak are encouraged to share on social media or review platforms to foster a supportive community where students can connect and encourage each other.

While successfully presenting how students were satisfied with ESLA Speak and the factors affecting their satisfaction by statistical analysis with satisfactory reliability, this study may have some potential limitations. The first issue is the small sample size. Hence, the findings may not be representative of the wider population and should be interpreted with caution as preliminary insights rather than definitive conclusions. Future research should reconfirm these results in a much larger sample size and across more diverse populations. Another limitation is that the study has just considered three antecedents of students' satisfaction which consist of perceived ease of use, perceived usefulness, and confirmation. This may not encompass all relevant variables. Further researches are needed to explore the effects of other variables on learners' satisfaction with ELSA Speak.

## 7. Conclusion

Pronunciation is one of the most important elements for learners to successfully communicate in English. This research paper has shed light on the extent to which English major students were satisfied with ELSA Speak to improve pronunciation as well as factors affecting learners' satisfaction. The results confirmed the great extent of participants' satisfaction and suggested the significant impact of learners' perceived ease of use and confirmation on satisfaction. Besides practical implications for curriculum designers, educators, and learners in using ELSA Speak in pronunciation classes, these findings could serve as a base for future research on learners' satisfaction with technology-enhanced language learning tools to foster the effectiveness of learning pronunciation.

## Supporting information

**S1 File. Questionnaire survey.**
(DOCX)

## Acknowledgments

We sincerely thank the authors of the cited research and the survey respondents who contributed to this study.

## Author Contributions

**Conceptualization:** Vi Thi Tuong Pham, Anh Tuan Pham.

**Data curation:** Vi Thi Tuong Pham, Anh Tuan Pham.

**Formal analysis:** Vi Thi Tuong Pham, Anh Tuan Pham.

**Funding acquisition:** Vi Thi Tuong Pham, Anh Tuan Pham.

**Investigation:** Vi Thi Tuong Pham, Anh Tuan Pham.

**Methodology:** Vi Thi Tuong Pham, Anh Tuan Pham.

**Project administration:** Vi Thi Tuong Pham, Anh Tuan Pham.

**Resources:** Vi Thi Tuong Pham, Anh Tuan Pham.

**Software:** Vi Thi Tuong Pham, Anh Tuan Pham.

**Supervision:** Vi Thi Tuong Pham, Anh Tuan Pham.

**Validation:** Vi Thi Tuong Pham, Anh Tuan Pham.

**Visualization:** Vi Thi Tuong Pham, Anh Tuan Pham.

**Writing – original draft:** Vi Thi Tuong Pham, Anh Tuan Pham.

**Writing – review & editing:** Vi Thi Tuong Pham, Anh Tuan Pham.

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
