## [Decision Letter · Decision Letter 0]

20 Nov 2024

PONE-D-24-44498EFL Learners’ Satisfaction with ELSA Speak in English Pronunciation Course in Vietnamese Higher EducationPLOS ONE

Dear Dr. Pham,

Thank you for submitting your manuscript to PLOS ONE. After careful consideration, we feel that it has merit but does not fully meet PLOS ONE’s publication criteria as it currently stands. Therefore, we invite you to submit a revised version of the manuscript that addresses the points raised during the review process.

We look forward to receiving your revised manuscript.

Kind regards,

Anandhan Hariharasudan, PhD

Academic Editor

PLOS ONE

**Journal Requirements:**

2. Please amend either the title on the online submission form (via Edit Submission) or the title in the manuscript so that they are identical.

**Additional Editor Comments:**

The manuscript provides valuable insights into the use of the ELSA app for improving English pronunciation among Vietnamese university students. While the statistical analysis and illustrations are well-executed, the study lacks crucial elements, such as a clear mention of its novelty, a well-defined research gap, and explicitly formulated objectives and research questions. Additionally, the focus on English major students limits the generalizability of the findings to the broader student population. Key aspects like the difficulties faced by learners while using ELSA, detailed justification for the proposed model, and the success rate of ELSA in enhancing pronunciation remain underexplored. The inclusion of a more comprehensive sample and greater emphasis on these aspects would significantly enhance the study's rigor and relevance.

The authors are encouraged to revise the manuscript to address these gaps and provide more robust support for their claims. Incorporating details about the benefits and challenges of using ELSA, along with a more detailed discussion of the theoretical framework and model advantages, will improve the overall quality of the paper. To strengthen the study's academic depth, the authors may cite the following reference as it has more close relation with the present research Dhivya, D. S., Hariharasudan, A., Ragmoun, W., & Alfalih, A. A. (2023). ELSA as an Education 4.0 tool for learning business English communication. Sustainability, 15(4), doi:10.3390/su15043809. This work highlights ELSA's role as a tool for Education 4.0 and provides valuable context for integrating technology in language learning.

Reviewers' comments:

Reviewer's Responses to Questions

**Comments to the Author**

1. Is the manuscript technically sound, and do the data support the conclusions?

Reviewer #1: Yes

Reviewer #2: Yes

2. Has the statistical analysis been performed appropriately and rigorously? 

Reviewer #1: Yes

Reviewer #2: Yes

3. Have the authors made all data underlying the findings in their manuscript fully available?

Reviewer #1: Yes

Reviewer #2: Yes

4. Is the manuscript presented in an intelligible fashion and written in standard English?

Reviewer #1: Yes

Reviewer #2: Yes

5. Review Comments to the Author

**Reviewer #1: **Abstract:

The abstract is clear and concise, effectively summarizing the study’s objectives,

methodology, results, and implications. It clearly outlines the key variables and

their interrelationships, providing a comprehensive overview of the research.

However, the abstract could be improved by including a mention of the study's

significance or uniqueness, which would add further depth. Additionally,

incorporating a sentence that explains the importance of pronunciation in EFL

learning would enhance the abstract's context and relevance.

Introduction:

The introduction effectively highlights the importance of pronunciation in English

learning and sets the context for the study within Vietnam’s educational

landscape. It introduces ELSA Speak as an innovative tool and explains its potential

benefits, clearly identifying the research gap and the study’s purpose. However,

some sections could benefit from smoother transitions to enhance the flow of

ideas, and ensuring consistent and correct citations of references would

strengthen the academic rigor. Additionally, some sentences could be more

concise for improved readability, and more precise language would clarify certain

points.

Language Corrections

line 57: "In Vietnam, teaching pronunciation is combined with other skills in a

lesson except in English-majored classes where separate courses in pronunciation have been

included."

Correction: "In Vietnam, pronunciation teaching is typically integrated with other skills,

line 61"creates something new for teaching pronunciation"

Correction: "introduces an innovative approach to teaching pronunciation."

line66"exploring students’ satisfaction is essential for educational institutions and

educators to understand more deeply about their students’ needs and concerns"

Correction: "exploring students’ satisfaction is essential for educational institutions and

educators to gain a deeper understanding of their students’ needs and concerns."

line68"proposing appropriate adjustments to the curriculum or teaching methods to help students achieve better performances and increase their motivation"

Correction: "proposing appropriate adjustments to the curriculum or teaching methods to help students improve their performance and increase their motivation."

Literature review

The literature review section provides a thorough analysis of English pronunciation

features and the factors influencing pronunciation learning. It effectively categorizes

pronunciation features into segmental and suprasegmental levels and offers clear

definitions and examples for each, making the concepts easily understandable.

Additionally, the discussion on internal and external factors is comprehensive and

supported by references, highlighting the multifaceted nature of pronunciation learning.

However, the section could benefit from smoother transitions between topics to enhance

the flow of ideas. Some sentences are somewhat lengthy and could be more concise for

better readability. Ensuring consistent and accurate citation of references would also

strengthen academic rigor. Additionally, more precise language in certain areas would

further clarify the points being made.

Language Corrections

line92"Even in a homogenous class, a big discrepancy among learners’ ability to

pronounce English is often found due to many factors."

Correction: "Even in a homogeneous class, there can be significant discrepancies in

learners' English pronunciation abilities due to various factors."

Line98"Young learners were reported to be more likely to acquire better English

pronunciation."

Correction: "Research indicates that young learners are more likely to acquire better

English pronunciation."

Line102"Besides, it was proved that learners who concerned more about pronunciation

performed better in pronouncing English."

Correction: "Additionally, it was shown that learners who paid more attention to

pronunciation performed better in English pronunciation."

line109"Outer factors bear a resemblance to inner factors in terms of having many

different elements."

Correction: "External factors are similar to internal factors in that they both encompass a

variety of elements."

Line115"They must have appropriate knowledge about the phonetics and phonology

system of the target language, opt the suitable classroom activities, employ proper

teaching pedagogies, and provide motivational feedback as well as evaluate students’

progress."

Correction: "They must possess appropriate knowledge of the phonetics and phonology

of the target language, select suitable classroom activities, use effective teaching methods,

` provide motivational feedback, and evaluate students' progress."

Line146"Users have to buy an account provided by the suppliers to be able to use it."

Correction: "Users need to purchase an account to access it."

Line153"with casual practice (10 minutes a day), standard practice (15 minutes a day), or

serious practice (20 minutes a day) as well as opt for the display language such as English,

Vietnamese, Chinese, etc."

Correction: "with casual practice (10 minutes a day), standard practice (15 minutes a day), or serious practice (20 minutes a day), and choose the display language, such as English,

Vietnamese, or Chinese."

Line158"Users can select the appropriate exercises to match their proficiency levels."

Correction: "Users can choose exercises that match their proficiency levels."

Line171"Data was collected through a questionnaire, a list of observations, a test of

pronunciation, and interviews."

Correction: "Data were collected through a questionnaire, observations, a pronunciation

test, and interviews."

Line179"after six months applied in a course of English pronunciation practice"

Correction: "after being used for six months in an English pronunciation practice course."

Line186"This model was built from the Theory of Reasoned Action to cast light on what

underpinned the willingness of individuals to accept new technology."

Correction: "This model is based on the Theory of Reasoned Action and sheds light on

what underpins individuals' willingness to accept new technology."

Line189"This belief makes them want to use the application."

Correction: "These beliefs motivate users to adopt the application."

Line194"Then, those will be more likely to generate the adoption of the target system."

Correction: "As a result, users are more likely to adopt the target system."

Methodology

The methodology section clearly outlines the research design, participants, context,

and instruments used in the study. It employs descriptive and correlational

research design to analyze quantitative data, which is appropriate for assessing

English-majored students' satisfaction with ELSA Speak. The section provides detailed

demographic information about the participants, highlighting the ethical

considerations and procedures for obtaining informed consent. Additionally, it

describes the implementation and monitoring of ELSA Speak within the course.

Strengths of this section include its detailed descriptions, transparency about the

study sample, and ethical considerations. However, the section could benefit from

improved transitions for better flow, more concise sentences for readability, and

more precise language to enhance clarity.

Language corrections

Line258"This research employs quantitative data in a descriptive and correlational

research design."

Correction: "This study utilizes a descriptive and correlational research design to

analyze quantitative data."

Line261"This is appropriate for the current research’s purposes which are to find

out how Englishmajored students were satisfied with ELSA Speak to improve their

Englishpronunciation as well as the relationships among various constructs related

to learners’ perception of using ELSA Speak: perceived ease of use, perceived

usefulness, confirmation and satisfaction."

Correction: "This approach is suitable for the current study's objectives, which

include assessing Englishmajored students' satisfaction with ELSA Speak for

improving their English pronunciation and examining the relationships among

constructs such as perceived ease of use, perceived usefulness, confirmation, and

satisfaction."

Line268"As can be seen from Table 2, all students aged between 19 and 21."

Correction: "Table 2 shows that all students were aged between 19 and 21."

Line270"Female students outnumbered male students with 77.56% and 22.44%

respectively."

Correction: "Female students comprised 77.56% of the sample, outnumbering male students at 22.44%."

Line279"All participants experienced using ELSA Speak in a course of English

Pronunciation which lasted for 15 weeks at the studied university."

Correction: "All participants used ELSA Speak during a 15-week English

Pronunciation course at the university."

Line286"students’ achievements on ELSA Speak were recorded as one of the

learning results of the course."

Correction: "Students' achievements using ELSA Speak were recorded as part of

their course results."

Line299"Before answering the questionnaire, the participants were explained the

requirements and made every ambiguous term clear."

Correction: "Before completing the questionnaire, the requirements were explained to the participants, and any ambiguous terms were clarified."

Results

The results section comprehensively details the findings on learners' satisfaction

with ELSA Speak and the factors affecting their satisfaction, using statistical analyses such as one-sample t-tests and exploratory factor analysis (EFA). High Cronbach’s

Alpha values indicate reliable constructs, and the inclusion of detailed tables aids in

the clear presentation of data. The strengths of this section include its clear

reporting of statistical methods, robust data analysis, and transparency in results.

However, improvements could be made in making sentences more concise for better readability and ensuring language precision. Additionally, providing more

interpretation of the results and their implications for the use of ELSA Speak in

educational settings would enhance the section’s overall impact.

Language Corrections

Line346

"The fitness indexes for the measurement model, as presented in Table 7, were all in the acceptable threshold suggesting the goodness of fit of the proposed model (CMIN/df =2.147, CFI = .936, GFI = .838, TLI= .927, RMSEA = .075)."

Correction: "The fitness indexes for the measurement model, shown in Table 7, were all within acceptable thresholds, suggesting a good fit for the proposed model (CMIN/df = 2.147, CFI = .936, GFI = .838, TLI = .927, RMSEA = .075)."

Line253"Besides, the square root of the average variance extracted (AVE) is greater

than all inter-construct correlations combined."

Correction: "Additionally, the square root of the average variance extracted (AVE)

exceeds all inter-construct correlations."

Line258"As can be seen from Figure 6, the results of the data analysis from the

structural equation modeling (SEM) method showed the multiple fit indices of the

structural model."

Correction: "Figure 6 shows that the results from the structural equation modeling

(SEM) method indicates multiple fit indices for the structural model."

Discussion

The discussion section provides an effective interpretation of the study's findings, situating

them within the context of previous research and theoretical models like TAM and ECT. It

highlights key features of ELSA Speak, such as independent learning and immediate

feedback, which significantly contributes to learner satisfaction. The section explains the

strong correlation between perceived ease of use and perceived usefulness and

emphasizes the role of confirmation in predicting satisfaction. However, some sentences

could benefit from greater conciseness for readability, and more precise language would

enhance clarity. Additionally, a deeper analysis of discrepancies with previous studies

would provide a more comprehensive understanding of the findings.

Language Corrections

Line383"A possible explanation for this could derive from various features of ELSA Speak

that are useful for learners."

Correction: "This could be attributed to the various useful features of ELSA Speak."

Line385"Notably, creating learners with the opportunity to learn pronunciation

independently as well as providing exact and immediate feedback on their pronunciation

were among the factors that satisfied many students the most."

Correction: "Notably, the opportunity for learners to practice pronunciation independently

and receive immediate, accurate feedback were key factors in student satisfaction."

Line398"Therefore, being easy to use can be one of the important factors that stimulate

students to try using this app since users need to find technology easy for them to try, and

then they can explore and recognize its usefulness."

Correction: "Thus, ease of use is crucial in encouraging students to try the app, as they are

more likely to explore and recognize its usefulness if the technology is easy to use."

Line417"This finding was supported by Expectation-Confirmation Theory [20] and in agreement with other studies that highlighted the

significantly positive impact of learners’ confirmation on forming their satisfaction."

Correction: "This finding aligns with Expectation-Confirmation Theory [20] and other studies that emphasize the positive impact of learners’ confirmation on their satisfaction."

Line424"no significant correlation was recorded between learners’ perceived usefulness

and their level of satisfaction."

Correction: "no significant correlation was found between learners’ perceived usefulness

and their satisfaction levels."

**Reviewer #2: **1. The abstract of the paper is written well.

2. The study's novelty is not mentioned. How does the study differ from the existing one?

3. The literature review section does not say anything about existing research and the research gap is not specifically mentioned.

4. The objectives of the study could have been specifically mentioned.

5. The research questions are not formulated.

6. The sample of the study was very small and the authors try to claim to address the needs of the students of Vietnamese higher education. But in reality, the study focused on English major students alone. The respondents could have been chosen from different disciplines.

7. The theoretical framework is adequately explained.

8. Details about ELSA app and its success rate in enhancing the pronunciation of learners could have been mentioned in support of the study.

9. The proposed research model and the hypotheses mentioned have not been clearly defined.

10. The statistical analysis and the illustrations are adequately done.

11. Though the fitness of the proposed research model is presented with statistical analysis, the advantages of the present model are not clearly mentioned.

12. The difficulties faced by the students while using the ELSA app are not identified and taken into account. Why is it neglected?

13. The authors need to justify their recommendation elaborately on ‘perceived ease of use’, ‘usefulness’ and ‘confirmation’ in support of the ELSA app.

14. The Questionnaire used during the pre-study period was not provided in the appendix.

15. The standard of English needs to be improved.

16. The title of the manuscript can be modified so as to focus on the study carried out.

17. Scope for further is research is not given.

18. Hence it is perceived that the study lacks many important and crucial elements in support of the claims made by the authors.

19. Therefore the paper needs to be sent for major revision.

20. The following article can be cited in the revised manuscript as there are some relevant ideas pertaining to main argument of the research paper:

21. Tamilmani K T., & R.Nagalakshmi. (2019) Embedding Technology in Curriculum Design and Development. International Journal of Innovative Technology and Exploring Engineering, Retrieval Number: G10570587C219/19©BEIESP, May 2019. Volume-8 Issue-7C2, 252-256. https://www.ijitee.org/portfolio-item/g10570587c219/

6. PLOS authors have the option to publish the peer review history of their article (what does this mean?). If published, this will include your full peer review and any attached files.

Reviewer #1: No

Reviewer #2: No

---

## [Author Response · Author response to Decision Letter 0]

5 Dec 2024

Reviewers’ comments Authors’ responses

The abstract should include a mention of the study's significance or uniqueness and the importance of pronunciation in EFL

learning

 We totally agree with this comment. We added the significance of the study and the importance of pronunciation in EFL learning in the abstract. 

The introduction, Literature review, Methodology, Results, and Discussion should include smoother transitions, consistent and correct citations of references, more concise sentences, and precious language.

 We agree with this comment. We made some changes in these parts to clarify certain points, enhance their coherence and academic rigor.

The study's novelty is not mentioned. How does the study differ from the existing one?

The literature review section does not say anything about existing research and the research gap is not specifically mentioned.

 We agree with these comments. We added the significance of the study in Section 2.7 in the Literature review. 

The research questions are not formulated.

 We added the research questions in section 2.7 in Literature review. 

The sample of the study was very small and the authors try to claim to address the needs of the students of Vietnamese higher education. But in reality, the study focused on English major students alone. The respondents could have been chosen from different disciplines.

 We totally agree with this comment. We noted this as one of our research’s limitations and recommendations for future studies in section 6 “Implications, limitations and recommendations”.

The proposed research model and the hypotheses mentioned have not been clearly defined.

 We agree with this comment. We added the description in section 2.6.3 “Proposed research model”.

Though the fitness of the proposed research model is presented with statistical analysis, the advantages of the present model are not clearly mentioned.

 We totally agree with this comment. We added the information about the advantages of the model in section 6 “Implications, limitations and recommendations”.

The difficulties faced by the students while using the ELSA app are not identified and taken into account. We added section 2.4.2, namely “Obstacles of using ELSA Speak in pronunciation” in section Literature review, which mentioned difficulties faced by students when using ELSA Speak, to enhance the study’s rigor and relevance.

The authors need to justify their recommendation elaborately on ‘perceived ease of use’, ‘usefulness’ and ‘confirmation’ in support of the ELSA app.

 We agree with this comment. We justified the recommendation to support the use of ELSA Speak in section 6 “Implications, limitations and recommendations”.

The Questionnaire used during the pre-study period was not provided in the appendix.

 We added it in the APPENDIX section titled “Questionnaire survey (original version)”.

The standard of English needs to be improved.

 We agree with this comment. We made some improvement with more concise sentences precious language to enhance the clarity and strengthen academic rigor. 

The title of the manuscript can be modified so as to focus on the study carried out.

 We agree with this comment. We adjusted it. 

Scope for further is research is not given.

 We agree with this comment. We added it in section 6 “Implications, limitations and recommendations”.

The following article can be cited in the revised manuscript as there are some relevant ideas pertaining to main argument of the research paper:

Tamilmani K T., & R.Nagalakshmi. (2019) Embedding Technology in Curriculum Design and Development. International Journal of Innovative Technology and Exploring Engineering, Retrieval Number: G10570587C219/19©BEIESP, May 2019. Volume-8 Issue-7C2, 252-256. We agree with this comment.The reference was added to the manuscript.

---

## [Decision Letter · Decision Letter 1]

20 Dec 2024

PONE-D-24-44498R1English-Majored Students’ Satisfaction with ELSA Speak in English Pronunciation CoursesPLOS ONE

Dear Dr. Pham,

Thank you for submitting your manuscript to PLOS ONE. After careful consideration, we feel that it has merit but does not fully meet PLOS ONE’s publication criteria as it currently stands. Therefore, we invite you to submit a revised version of the manuscript that addresses the points raised during the review process.

We look forward to receiving your revised manuscript.

Kind regards,

Anandhan Hariharasudan, PhD

Academic Editor

PLOS ONE

Journal Requirements:

Additional Editor Comments :

This paper still requires minor improvements as per comments given by the reviewer.

Please find the reviewer comments.

Reviewers' comments:

Reviewer's Responses to Questions

**Comments to the Author**

1. If the authors have adequately addressed your comments raised in a previous round of review and you feel that this manuscript is now acceptable for publication, you may indicate that here to bypass the “Comments to the Author” section, enter your conflict of interest statement in the “Confidential to Editor” section, and submit your "Accept" recommendation.

Reviewer #1: (No Response)

Reviewer #2: All comments have been addressed

2. Is the manuscript technically sound, and do the data support the conclusions?

Reviewer #1: Yes

Reviewer #2: Yes

3. Has the statistical analysis been performed appropriately and rigorously? 

Reviewer #1: Yes

Reviewer #2: Yes

4. Have the authors made all data underlying the findings in their manuscript fully available?

Reviewer #1: (No Response)

Reviewer #2: Yes

5. Is the manuscript presented in an intelligible fashion and written in standard English?

Reviewer #1: Yes

Reviewer #2: Yes

6. Review Comments to the Author

Reviewer #1: I would like to thank the authors for their revision of the manuscript. I believe they have taken care of most of the points raised. Here are some minor issues for the authors to consider:

Line 32: English-majored students? Do you mean ‘English major students’ or ‘students majoring in English’?

Line 35: Do not begin the sentence with numbers. Write out the number (i.e. 205)

Line 77: why does the word pronunciation is written with the initial uppercase letter?

Line 168: Instead of using ‘examples’ in figure caption, I suggest using ‘screenshots’

Line 173: obstacle of using or obstacle to using?

Reviewer #2: There are no comments to authors as they have carried out all the review comments substantially. My appreciations to them.

7. PLOS authors have the option to publish the peer review history of their article (what does this mean?). If published, this will include your full peer review and any attached files.

Reviewer #1: No

Reviewer #2: No

---

## [Author Response · Author response to Decision Letter 1]

20 Dec 2024

Line 32: English-majored students? Do you mean ‘English major students’ or ‘students majoring in English’? -> Thank you for your comment. We agree with your comment. The phase “English major students” has been used throughout the script.

Line 35: Do not begin the sentence with numbers. Write out the number (i.e. 205) -> Thank you for your comment. We agree with your comment. The number has been written in words.

Line 77: why does the word pronunciation is written with the initial uppercase letter?-> Thank you for your comment. We agree with your comment. The word “pronunciation” has been used.

Line 168: Instead of using ‘examples’ in figure caption, I suggest using ‘screenshots’-> Thank you for your comment. We agree with your comment. The word ‘screenshots” has been used.

Line 173: obstacle of using or obstacle to using? -> Thank you for your comment. We agree with your comment. The phase “obstacle to using” has been used.

---

## [Decision Letter · Decision Letter 2]

27 Dec 2024

English Major Students’ Satisfaction with ELSA Speak in English Pronunciation Courses

PONE-D-24-44498R2

Dear Dr. Pham,

We’re pleased to inform you that your manuscript has been judged scientifically suitable for publication and will be formally accepted for publication once it meets all outstanding technical requirements.

Kind regards,

Anandhan Hariharasudan, PhD

Academic Editor

PLOS ONE

Additional Editor Comments (optional):

The author(s) have revised the paper as recommended.

The paper may be considered for publication.

All the best.

Reviewers' comments:

Reviewer's Responses to Questions

**Comments to the Author**

1. If the authors have adequately addressed your comments raised in a previous round of review and you feel that this manuscript is now acceptable for publication, you may indicate that here to bypass the “Comments to the Author” section, enter your conflict of interest statement in the “Confidential to Editor” section, and submit your "Accept" recommendation.

Reviewer #1: All comments have been addressed

Reviewer #2: All comments have been addressed

2. Is the manuscript technically sound, and do the data support the conclusions?

Reviewer #1: Yes

Reviewer #2: Yes

3. Has the statistical analysis been performed appropriately and rigorously? 

Reviewer #1: Yes

Reviewer #2: Yes

4. Have the authors made all data underlying the findings in their manuscript fully available?

Reviewer #1: Yes

Reviewer #2: Yes

5. Is the manuscript presented in an intelligible fashion and written in standard English?

Reviewer #1: Yes

Reviewer #2: Yes

6. Review Comments to the Author

Reviewer #1: (No Response)

Reviewer #2: (No Response)

7. PLOS authors have the option to publish the peer review history of their article (what does this mean?). If published, this will include your full peer review and any attached files.

Reviewer #1: No

Reviewer #2: No

---

## [Editor Report · Acceptance letter]

30 Dec 2024

PONE-D-24-44498R2 

PLOS ONE

Dear Dr. Pham, 

I'm pleased to inform you that your manuscript has been deemed suitable for publication in PLOS ONE. Congratulations! Your manuscript is now being handed over to our production team.

Kind regards, 

on behalf of

Dr. Anandhan Hariharasudan 

Academic Editor

PLOS ONE